# Nanoencapsulation of amitriptyline enhances the potency of antidepressant-like effects and exhibits anxiolytic-like effects in Wistar rats

Ramón Eduardo Valadez-Lemus[1], José L. Góngora-Alfaro[2], Juana María Jiménez-Vargas[1,3], Javier Alamilla[4,5]*, Néstor Mendoza-Muñoz[1]*

**1** Facultad de Ciencias Químicas, Universidad de Colima, Colima, México, **2** Centro de Investigaciones Regionales "Dr. Hideyo Noguchi", Universidad Autónoma de Yucatán, Yucatán, México, **3** Consejo Nacional de Humanidades Ciencia y Tecnología (CONAHCYT), México City, México, **4** Centro Universitario de Investigaciones Biomédicas, Universidad de Colima, Colima, México, **5** Investigador por México-CONAHCYT-Universidad de Colima, Colima, México

* alamilla78@gmail.com, jfalamillago@conahcyt.mx (JA); nmendoza0@ucol.mx (NM-M)

## Abstract

Depression poses a significant global health challenge, affecting an estimated 300 million people worldwide. While amitriptyline (Ami) remains one of the most effective antidepressants, its numerous side-effects contribute to a high dropout rate among patients. Addressing this issue requires exploring methods to enhance its bioavailability and reduce dosage. In this study, we describe a technique for producing amitriptyline nanoparticles (Ami-NPs) to improve the drug's efficiency. The effectiveness was assessed by comparing the dose-response curves of Ami-NPs and non-encapsulated Ami in male and female Wistar rats subjected to the forced swimming test (FST). Ami-NPs were fabricated using nanoprecipitation, with a copolymer of poly (methyl vinyl ether/maleic acid) as the encapsulant, and a 3% solution of poloxamer F-127 as surfactant stabilizer. A Box-Behnken design was used to optimize the production of Ami-NPs, resulting in nanoparticles with the following optimal characteristics: a size of 198.6 ± 38.1 nm, a polydispersity index of 0.005 ± 0.03 nm, a zeta potential of -32 ± 6 mV, and encapsulation efficiency of 79.1 ± 7.4%. Ami-NPs showed higher potency and efficacy in reducing immobility during the FST ($ED_{50}$ = 7.06 mg/kg, $E_{max}$ = 41.1%), compared to amitriptyline in solution (Ami-S) ($ED_{50}$ = 11.89 mg/kg, $E_{max}$ = 33.2%). The $E_{max}$ of Ami-NPs occurred at 12 mg/kg, while Ami-S peaked at 15.8 mg/kg. In the open field test, only treatment with Ami-NPs (12 mg/kg) and the empty nanoparticles increased immobility. In the elevated plus-maze, treatment with Ami-NPs (12 mg/kg) significantly reduced closed-arm entries (2.1 ± 0.6), compared to control solution (9.5 ± 1.8), control nanoparticles (8 ± 1.0) and Ami-S (11.5 ± 2). In the marble burying test, Ami-NPs (12 mg/kg) significantly reduced buried marbles (2.4 ± 0.4) compared to control nanoparticles (8.7 ± 1.2). These findings suggest that Ami-NPs could be a promising approach to enhance Ami bioavailability, thereby increasing its potency and antidepressant efficacy, while improving anxiolytic-like effects.

**Data availability statement:** All relevant data are within the manuscript and its Supporting Information files.

**Funding:** This work was supported by the Consejo Nacional de Ciencia y Tecnología-Secretaría de Educación Pública (CONACYT-SEP) under Grant CB A1-S-39237 to NMM, and by Consejo Nacional de Humanidades, Ciencia y Tecnología, Ciencia Básica y de Frontera (CONAHCYT- CBF) 2023-2024-3517 to JA. The funders had no role in study design, data collection and analysis, decision to publish, or preparation of the manuscript.

**Competing interests:** The authors have declared that no competing interests exist.

## Introduction

Major depression is a mood disorder marked by persistent sadness, anhedonia (lack of interest in daily activities), difficulty concentrating, and, in severe cases, suicidal tendencies. Globally, around 300 million people suffer major depression, making it the most prevalent disabling illness [1]. The COVID-19 pandemic has further exacerbated the prevalence of anxiety and depression disorders, with a 25% increase worldwide [2].

The primary treatment for major depression involves a combination of antidepressant medications and cognitive-behavioral therapy [3]. Among various antidepressants, the tricyclic antidepressant amitriptyline (Ami) is particularly effective, acting by inhibiting the reuptake of serotonin and norepinephrine [4,5]. In addition, it is also used as an analgesic to treat neuropathic pain and neuralgia [6–8]. However, its numerous side effects, including drowsiness, and cardiovascular effects, between others, limit patient adherence [9,10].

Nanometric drug delivery systems have emerged as promising pharmacological options, enabling targeted drug delivery and controlled release over time. Nanoparticles also facilitate transport of drugs to the brain by interacting with the blood brain barrier (BBB) [11]. These properties make nanoparticle an excellent delivery system for psychotropic drugs, enhancing their beneficial effects while overcoming the limitations of traditional formulations. Several studies highlight the advantages of nanoparticle formulations of antidepressant drugs, showing improved antidepressant-like properties [12–16]. In this context, we propose that nanoencapsulation could to potentially overcome the limited bioavailability of Ami, which is hindered by p-glycoprotein, an ABCB transporter expressed by the BBB cells [17]. Various nanocarriers have been proposed to overcome this barrier. It is important to note that factors such as size, shape, biodegradability, bioadhesion, lipophilicity, surface charge, and surface modification influence peripheral metabolism and ultimately determine the amount of drug delivery to the brain [18]. Polymeric nanocapsules, in particular, offer the ability to modulate physicochemical properties that facilitate drug passage across the BBB. Additionally, nanoencapsulation, has been shown to efficiently protect against degradation from temperature, pH fluctuations, light exposure, enzyme activity, and support high drug loading capacity and biocompatibility [19].

Nanoencapsulation could also addresses the issue of high protein and tissue biding of Ami (approximately 95%), which limits access to the Central Nervous System (CNS) and reduces the antidepressant effect [5,20,21]. The evidence indicates that nanoencapsulation helps bypass active transport systems and enhances cerebral bioavailability. Incorporating drugs into nanoparticles formed with various copolymers and surfactants aids adherence and passage though BBB cell layers [22,23].

A previous study reported the development of Ami nanoparticles (Ami-NPs) using polyethylene glycol-poly lactic acid-co-glycolic acid copolymer was reported, which demonstrated accelerated wound healing in diabetic rats [24]. Other studies have shown methods for preparing Ami-NPs using spray freeze drying [25] and ultrasonic spray-assisted electrostatic adsorption [26]. However, there is a lack of research evaluating whether nanoencapsulation of Ami can improve its antidepressant-like effects through animal behavioral tests.

In this study, we report the preparation and optimization of Ami-NPs, focusing on particle size, zeta potential and encapsulation efficiency to facilitate passage through lipid bilayers including the BBB [23,27,28]. A Box-Behnken design (BBD) was used to test various proportions of Ami and copolymers. Poly [methyl vinyl ether/maleic acid] (PMVEMA, Gantrez S-97®), was selected as first copolymer due to its demonstrated efficacy in nanoparticle formation and its adhesive properties for drug transport through mucosal walls [29]. The second copolymer used was the nonionic surfactant Poloxamer 407 (Pluronic® F127), which has a hydrophobic core of polypropylene oxide flanked by two hydrophilic ethylene oxide chains

(ABA-type) [30]. The incorporation of Poloxamer 407 into poly (lactic-co-glycolic acid)-poly (ethylene glycol) copolymer nanoparticles has been shown to enhance brain penetration [31]. Poloxamer 407 has also been successfully used to solubilize hydrophobic drugs and develop prolonged-release formulations for intramuscular and intraperitoneal administration in pre-clinical studies [32,33].

This investigation had two aims: to describe the method used to develop and character-ize Ami nanoparticles and to evaluate their antidepressant- and anxiolytic-like properties using behavioral tests in rats. To evaluate whether Ami-NPs enhance antidepressant potency and efficacy, dose-response curves (DRC) were generated for both Ami-NPs and Ami in solution (Ami-S) to compare their median effective dose ($ED_{50}$) and maximum effects ($E_{max}$) in reducing immobility time in the forced swimming test (FST), a widely used preclinical model for identifying antidepressant drugs [34]. The main finding of this study indicates that the method used successfully produced Ami nanoparticles, which displayed enhanced antidepressant-like potency and possessed anxiolytic characteristics.

## Materials and methods

### Materials and chemicals

The copolymer Poly (MetilVinilEther/Maleic Acid) (PMVE/MA)(Gantrez S-97®) was generously provided by Ashland (Mexico). Calcium carbonate (> 99.0%) was obtained from Golden Bell (Mexico), and Poloxamer 407 was purchased from Sigma Aldrich (USA). Ethanol (99%) was obtained from JT Baker, and Ami hydrochloride (98-99%) was kindly donated by Neolpharma S.A. de C.V. (Mexico). Methanol (99.99%), acetonitrile (99.9%), and HPLC grade water (99.9%) for subsequent HPLC analysis were also obtained from Fermont and JT Baker, respectively.

### Preparation of amitriptyline polymeric nanoparticles

Prior to nanoparticle fabrication, Ami hydrochloride was neutralized to its base form to enhance encapsulation efficiency. For this, a 5% $NaHCO_3$ solution was prepared and added dropwise to a 1 mg/mL solution of Ami hydrochloride. The resulting Ami base was then extracted using ethyl acetate and evaporated to dryness, yielding a yellow oil corresponding to Ami free base, all experiments were conducted using this Ami base form.

Ami-NPs were prepared using an adapted nanoprecipitation method based on Fessi [35]. Typically, 50 mg of PMVE/MA and 30 mg of Ami free base were dissolved in 30 mL of acetone (organic phase). This solution was then added dropwise to a 50 mL aqueous solution of Polox-amer 407 (3%) and ethanol (1:1 v:v). The resultant dispersion was stirred on a magnetic plate at 500 to 700 rpm at room temperature and then evaporated under reduced pressure at 40 °C. The Ami-NPs suspension was filtered using a tangential filter (Sartorius, Germany) with a molecular weight cut off (MWCO) of 100 kDa and recirculated using a peristaltic pump. The final nanoparticle suspension was adjusted to 25 mL, and subsequently lyophilized, and stored in a desiccator until use.

### Optimization of the particle size of the Ami-NPs by design of experiments

A Box-Behenken design (BBD) was selected to optimize the particle size of Ami-NPs, with particle size as the primary response variable. Optimization was aimed at minimizing the hydrodynamic diameter under specific preparation conditions. A total of 15 experiments were conducted, including 12 factorial points at the midpoints of the edges of the process space and 3 replicates at the center point for estimating the pure error sum of squares. This approach enabled the selection of the best model among the linear, two-factor interaction model, and

quadratic models. The factors evaluated for the BBD are described in Table 1. The following second-order polynomial equation was used to interpret the results, considering the coefficient magnitude and mathematical sign (i.e., positive or negative).

$$Y = \beta_0 + \beta_1 A + \beta_2 B + \beta_3 C + \beta_{11} A^2 + \beta_{22} B^2 + \beta_{33} C^2 + \beta_{12} AB + \beta_{13} AC + \beta_{23} BC$$

where Y is the predicted response (particle size), $\beta_0$ is the intercept, $\beta_1$, $\beta_2$, and $\beta_3$, are linear coefficients, $\beta_{11}$, $\beta_{22}$, and $\beta_{33}$ are squared coefficients and quadratic term, $\beta_{12}$, $\beta_{13}$, and $\beta_{23}$ are interaction coefficients, and A, B, and C are independent variables.

## Characterization of optimized Ami-NPs

**Particle size, size distribution and zeta potential.** Particle size, represented as the hydrodynamic diameter, and polydispersity index (PDI) were determined using Dynamic Light Scattering (DLS). The surface charge, expressed as zeta potential, was measured by laser Doppler anemometry, which is based on the electrophoretic mobility of particles in an electrical field. The analysis was performed using a dynamic laser diffractometer (Brookhaven Instruments Corporation, USA) equipped with He/Ne laser operating at a wavelength of 678 nm and a potency of 10 mW. Measurements were taken at a fixed angle of 90° for 120 seconds, with ultrapure water at 25°C as the dispersion medium. Each system was measured in triplicate.

**Morphology.** To assess the shape and confirm the size of Ami-NPs, Scanning Electron Microscopy (SEM) was performed using an electronic microscope (Jeol, model JSM-6390, USA). A drop of the Ami-NPs dispersion was placed on a 100 nm polyvinylidene fluoride membrane as support. After drying, the membrane containing Ami-NPs was coated with a layer of ≈ 2 nm of gold by sputtering. The Ami-NPs were observed under low vacuum mode using an accelerating voltage of 15 kV.

**Amitriptyline encapsulation efficiency.** To determine the encapsulation efficiency (EE%), 5 mL of Ami-NPs suspension were frozen and then lyophilized for 24 hours to remove water. The amount of Ami was then quantified by HPLC using a modified methodology of Sevak *et al.* [36]. The HPLC analysis was conducted on an Alliance model e2695 (Waters, USA) separation module, which includes a quaternary pump, online vacuum degasser, autosampler, and column oven, coupled with a diode array detector (DAD) model 2998 with a wavelength range of 190 to 800 nm. For quantification, 20 mg of lyophilized Ami-NPs were weighted, suspended in methanol, sonicated for 5 minutes, filtered with 0.23 μm membrane, and adjusted to 10 mL with methanol. Then, 20 μl of the solution was injected, and separation was performed using a linear gradient elution with methanol and 20 mM phosphate buffer (pH = 3), ranging from 20 to 80% of organic phase (v/ v). A C18 column (3.5 μm, 4.6 mm x 150 mm) from X-Bridge (Ireland) was employed at a flow rate of 0.6 mL/min, at room temperature. Gradient mobile phases were prepared using the pump, and the analysis time was approximately 30 minutes. The DAD was set to record at nm.

**Table 1. Evaluated factors for the elaboration of Ami-NPs in the BBD, in response to particle size.**

| Factor | Name | Minimum | Average | Maximum | Response |
|--------|------|---------|---------|---------|----------|
| A | PMVE/MA concentration | 50 mg | 100 mg | 150 mg | Particle size |
| B | Amitriptyline concentration | 1:3 of copolymer concentration | 1:2 of copolymer concentration | 1:1 of copolymer concentration | |
| C | Volume of acetone | 1:3 of aqueous phase | 1:2 of aqueous phase | 1:1 of aqueous phase | |

The EE% of Ami in the Ami-NPs was calculated using the following equation:

$$EE\% = \left( \frac{Amount\ of\ Ami\ quantifyeds\ in\ the\ formulation}{Amount\ of\ Ami\ expected\ in\ formulation} \right) 100\%$$

The study was conducted in quadruplicate, and the mean values with standard deviations were reported. The analytical method validation followed International Conference on Harmonisation guidelines [37], reporting parameter such as linearity, limits of detection (LoD), and quantification (LoQ).

**In vitro release assays.** For this analysis, 10 mL of a standardized Ami-NPs dispersion was placed in a cellulose membrane dialysis bag (12400 MWCO) (Sigma Aldrich, USA). The dialysis bag was immersed in 500 mL of an aqueous medium of PBS 1 M (pH 7.4) to simulate plasma pH conditions, with sink conditions maintained throughout to ensure sufficient media for the drug dissolution. The first aliquot (1 mL) was collected 30 min after the start of the assay; followed by hourly collections during 8 h, and the last sample was taken 24 h after starting the test to evaluate the continuous release behavior of the formulation. The collected samples were filtered and analyzed using a UV-Visible spectrophotometer (Evolution 300 Thermo Scientific EV3, USA) at 240 nm for Ami quantification. A calibration curve was previously established.

## Ethics and behavioral animal tests

All procedures were approved by the Institutional Bioethics Committee of the University Center for Biomedical Research of the University of Colima in Mexico and were conducted in accordance with the National Institutes of Health Guide for Care and Use of Laboratory Animals (USA, NIH Publications No. 80 –23) revised in 1996. Behavioral tests were performed as described by Reyes-Mendez *et al.* [38], following the protocols of Detke y Lucki *et al.*, [39]. Adult young male and female Wistar rats (200-250 g) were used for behavioral experiments. Animals were group-housed in polycarbonate cages (length 29.5 cm, width 19 cm, height 15 cm). One week prior to the experiments, the rats were individually housed in a sound-attenuated room with controlled temperature (23 ± 1 °C, 40–50% humidity) under 12:12 light-dark cycle (lights on at 07:00). Food and water were available *ad libitum*. All necessary measures were taken to minimize the number of animals used in this study and alleviate their suffering. To acclimate the animals and reduce stress, the experimenter handled the rats daily for one week before the behavioral tests, which were conducted in an isolated and soundproof room. The animals were transferred to the experimental room one hour before the start of the experiments. Behavioral tests were video recorded and later analyzed by two independent observers blinded to the treatments administered to the rats. At the end of the experiments, rats were deeply anesthetized with pentobarbital (125 mg/kg) and, once the foot withdrawal reflex was absent, they were euthanized by cervical dislocation.

## Determination of estrous cycle phase

Female rats were tested during phases of low hormonal production (estrous-metestrus) [40]. The estrous cycle phase was determined by inspecting smears of vaginal washings made with 200 μL of sterile distilled water applied with a sterile micropipette tip inserted of 5-10 mm into the vagina. The samples were collected approximately at the same hour in the morning to reduce variability. The vaginal wash was spread on a slide and allowed to dry without fixation or staining. The smear was then photographed with a 40× objective magnification using an inverted microscope (Nikon-TMS100). Three independent observers classified the

cells from the vaginal smears to determine the estrous cycle phase based on the criteria of Cora *et al.* [40]. The estrous cycle phases were identified by their characteristic cytology: 1) Proestrus, characterized by a high number of small, nucleated epithelial cells with a relative uniform appearance. 2) Estrous, determined by a high number of anucleated epithelial cells. 3) Metestrus, identified by the presence of both anucleated and keratinized epithelial cells, along with neutrophils. 4) Diestrus, marked by a significant reduction in the number of anucleated keratinized epithelial cells, with moderated to low number of cells, including a combination of neutrophils, and small and large nucleated epithelial cells.

## Description of the FST, doses of Ami-S and Ami-NPs used, and control solutions

This test comprised two sessions: a conditioning pretest lasting 15 min, followed by the FST proper, conducted 24 hours later, lasting only 5 min [34,39,41]. All drug regimens were administered intraperitoneally at 23, 4, and 1 h before the FST, following the protocol outlined by Porsolt *et al.* [41]. The rat was placed inside a plastic cylinder (50 cm of height, 30 cm of diameter) filled with water (23 ± 1°C) up to 30 cm, as detailed by Detke *et al.* [39]. After each session, the animals were dried with a towel and placed under a heating lamp for 15 min to facilitate recovery from hypothermia. At the end of each trial, the water in the cylinder was discarded and replaced with fresh water for the subsequent test with another animal. Four ascending doses were administered in both experimental groups, along with their respective controls. The common doses were (in mg/kg): 4, 7, 12 and 17. Additionally, doses of 22 mg/kg and 10 mg/kg were administered in the Ami-S and Ami-NPs groups, respectively; to construct a comprehensive dose-response curve for each compound. The rationale behind using a wide range of doses was to thoroughly evaluate the antidepressant-like properties of the compounds as the doses increased. Control animals for the Ami-S group received only distilled water; controls for the Ami-NPs group were treated with a quantity of nanoparticles (control nanoparticles) equivalent to that contained in the dose of 12 mg/Kg of Ami-NPs, which produced the maximal effect in the FST. All drug doses were applied in a volume of 1 mL/kg.

## Open field test

Animals were randomly assigned to one of the following treatments: 1) Control solution; 2) control nanoparticles; 3) Ami (12 mg/kg); or 4) Ami-NPs (12 mg/kg). Each group consisted of 6 rats. In the open field test (OFT) experiments, a translucent acrylic box with a floor area of 50 × 50 cm and walls 40 cm in height was used. The floor was divided into 25 squares (10 cm × 10 cm, each). A black spot of approximately 2 cm in diameter was marked on the upper back of the animal using a permanent black ink marker. The rats were placed in the center of the box for 15 minutes, and the session was videotaped for later analysis, with focus on the first 10 minutes. Observers evaluated the following parameters: number of squares crossed by the black mark, total time of immobility (defined as the time that the animal remains still without moving its tail or paws, excluding head movements), the time the animal spent in the internal zone (the nine central squares of the floor) and external zone (squares along the box walls), number of feces, and frequency of stereotyped behaviors (rearing and self-grooming), as previously reported [42].

## Elevated plus maze

The drug treatments were the same as those used for the OFT. The elevated plus maze (EPM) with a height of 50 cm, was conducted from a black acrylic and consisted of two open arms (50 × 10 cm) and two enclosed arms (50 × 10 × 40 cm). To aid observer analysis, white lines

(3 mm in width) were drawn between the neutral zone (NZ) and each arm. Rats were placed in the NZ, facing the open arm opposite the experimenter. The animals were videotaped on the EPM for 10 minutes for offline analysis. Time spent in the open or closed arms was recorded each time the rat entered an arm with all, while time in the NZ was recorded when the rat placed at least one paw in this zone. The parameters measures examined for each experiment included: time (s) spent in open arms, closed arms, and neutral zone; number of open and closed arm entries; and total number of entries, as previously reported [42].

### Marble burying test

Animals received similar injection schedules as reported in the FST. Groups evaluated in this test were the same as those in the OFT and EPM. The marble burying test (MBT) was conducted in a clean Plexiglas cage ($27 \times 37 \times 15$ cm) containing a fresh 5 cm-thick layer of wood bedding. On top of the bedding, 20 glass marbles (15 mm diameter) were symmetrically arranged in 5 rows of 4 marbles each, as described by Kedia and Chattarji [43]. The test lasted 20 minutes, during which the animal was placed in the cage. Afterward, the animal was removed, and the cage was photographed. A marble was considered 'buried' is at least two-thirds of it was covered with bedding.

### Data analysis

All values were reported as the mean $\pm$ standard error of the mean (S.E.M.) (n = number of rats). Statistical analysis was performed using Prism 6 (GraphPad Software, La Jolla, CA, USA). Dose-effect relationships were evaluated using a sigmoidal dose-response equation: Y = Bottom + (Top − Bottom)/1 + 10LogED50 − X. One-way analysis of variance (ANOVA) followed by Tukey's post-hoc test was used, since Bartlett's test indicated homogeneity of variances. For the comparison of two groups, an independent $t$-test was applied. A two-tailed probability value of $p < 0.05$ was considered statistically significant.

## Results

### Preparation and characterization of Ami-NPs

The variation of three factors (acetone volume, Ami amount, and PMVE/MA amount) resulted in different particle sizes, as shown in Table 2. Notably, experimental run 7 exhibited the smallest average particle size (214.7 nm $\pm$ 29.1). In this experiment, a lower ratio Ami: PMVE/MA (25:50) and a higher volume of organic phase (30 mL of acetone) were used. Conversely, runs experiment number 4 and 14 showed the largest particle size (1165.1 $\pm$ 663.9 nm and 1002.7 $\pm$ 124.7 nm respectively). These experiments involved a higher amount of Ami and a lower volume of organic phase (Fig 1). The data were analyzed to derive a second-order polynomial equation correlating particle size (Y) with the proportions of their components:

$$Y = 637.0 - 96.5A + 262.3B + 95.9C - 53.6A^2 + 125.8B^2 - 96.9C^2$$
$$- 64.1AB + 71.2AC + 20.8BC$$

Using this equation, it was determined that to achieve Ami-NPs with a size of 200 nm, a mixture of 30 mL of the organic phase, 50 mg of copolymer (PMVE/MA), and 30 mg of Ami was required. This combination yielded Ami-NPs with an average size of 198.6 $\pm$ 38.1 nm and a PDI of 0.005 $\pm$ 0.03. Scanning electron microscopy revealed that optimized Ami-NPs have a spherical shape and a nanometric size (Fig 2). The stability of Ami-NPs, assessed via their effective electric charge, showed a zeta potential of -32 $\pm$ 4.8 mV. The suspension remained stable for 3 months without aggregation or significant changes in zeta potential.

**Table 2. Particle size and Polydispersity Index (PDI) obtained by variations in factors A, B, and C using the BBD.**

| Run | Acetone (mL) | Amitriptyline (mg) | PMVE/AM (mg) | Particle size (nm) | Polydispersity index (PDI) |
|-----|-----|-----|-----|-----|-----|
| 1 | 10 | 50 | 100 | 741.8 ± 34.1 | 0.210 ± 0.05 |
| 2 | 6.66 | 33.3 | 100 | 424.5 ± 37.2 | 0.215 ± 0.03 |
| 3 | 20 | 33.3 | 100 | 381.4 ± 119.9 | 0.212 ± 0.05 |
| 4 | 6.66 | 100 | 100 | 1165.1 ± 663.9 | 0.265 ± 0.03 |
| 5 | 20 | 100 | 100 | 865.7 ± 383.6 | 0.269 ± 0.12 |
| 6 | 6.66 | 25 | 50 | 571.6 ± 214.2 | 0.231 ± 0.07 |
| 7 | 20 | 25 | 50 | 214.7 ± 29.1 | 0.191 ± 0.10 |
| 8 | 10 | 50 | 100 | 654.33 ± 85.31 | 0.220 ± 0.095 |
| 9 | 6.66 | 75 | 150 | 615.9 ± 167.1 | 0.267 ± 0.003 |
| 10 | 20 | 75 | 150 | 543.8 ± 20.1 | 0.208 ± 0.05 |
| 11 | 10 | 16.6 | 50 | 369.9 ± 61.1 | 0.163 ± 0.05 |
| 12 | 10 | 50 | 50 | 764.9 ± 11.8 | 0.249 ± 0.04 |
| 13 | 10 | 50 | 150 | 525.2 ± 4.16 | 0.142 ± 0.11 |
| 14 | 10 | 150 | 150 | 1002.7 ± 124.7 | 0.122 ± 0.11 |
| 15 | 10 | 50 | 100 | 514.93 ± 172.40 | 0.197 ± 0.062 |

## Amitriptyline encapsulation efficiency

The analysis of encapsulation efficiency of Ami in the Ami-NPs was analyzed using HPLC. The retention times were 25 min for Ami, 4 min for PMVE/AM, and 5 min for Poloxamer 407, ensuring no coelution of components in the Ami-NPs. A calibration curve was constructed by plotting the area under the curve for Ami (HPLC) against known concentrations (mg/mL) of Ami. The calibration curve showed excellent linearity with an $R^2$ = 0.99, a LoD = 0.01 mg/mL, and a LoQ = 0.04 mg/mL. The concentration of encapsulated Ami in the Ami-NPs displayed an average encapsulation efficiency of 79.1 ± 7.4%.

## In vitro release studies

The in vitro release studies of Ami-NPs showed a release rate of approximately 78.0%, at 8 hours (Fig 3A). The release kinetics of Ami-NPs was best described by the Korsmeyer-Peppas model, as detailed in Table 3 and illustrated in Fig 3B.

## Forced swimming test

To investigate potential gender-based differences in response to a maximally effective dose of Ami (17 mg/kg), male and female rats (in estrous/metestrus) were subjected to the FST. As shown in Fig 4, no significant difference was observed in immobility times between male (148.2 ± 8.9 s) and female (160 ± 17.9 s) rats treated with this dose of Ami-S ($t_8$ = 0.6; $p$ = 0.5).

Next, to evaluate whether nanoencapsulation of the antidepressant affected its efficacy in reducing rat immobility in the FST, dose-response curves were generated for Ami-S and Ami-NPs. The doses tested were 4, 7, 12, 17, and 22 mg/kg for Ami-S and 4, 7, 10, 12 y 17 mg/Kg for Ami-NPs. Consistent with previous studies, Ami-S showed a significant dose-dependent reduction in immobility ($F_{[5, 27]}$ = 5.21, $p$ = 0.0018, one-way ANOVA), with an effective dose 50 (ED50) of 11.9 mg/kg and a maximum effect (Emax) of 33.3%, compared to the vehicle-treated control group. Similarly, Ami-NPs displayed a dose-dependent reduction in immobility ($F_{[5, 26]}$ = 5.65, $p$ = 0.0012, one-way ANOVA), with a reduced ED50 of 7.1 mg/kg and an increased Emax of 41.1% (Fig 5; Supporting information file (S1 File).

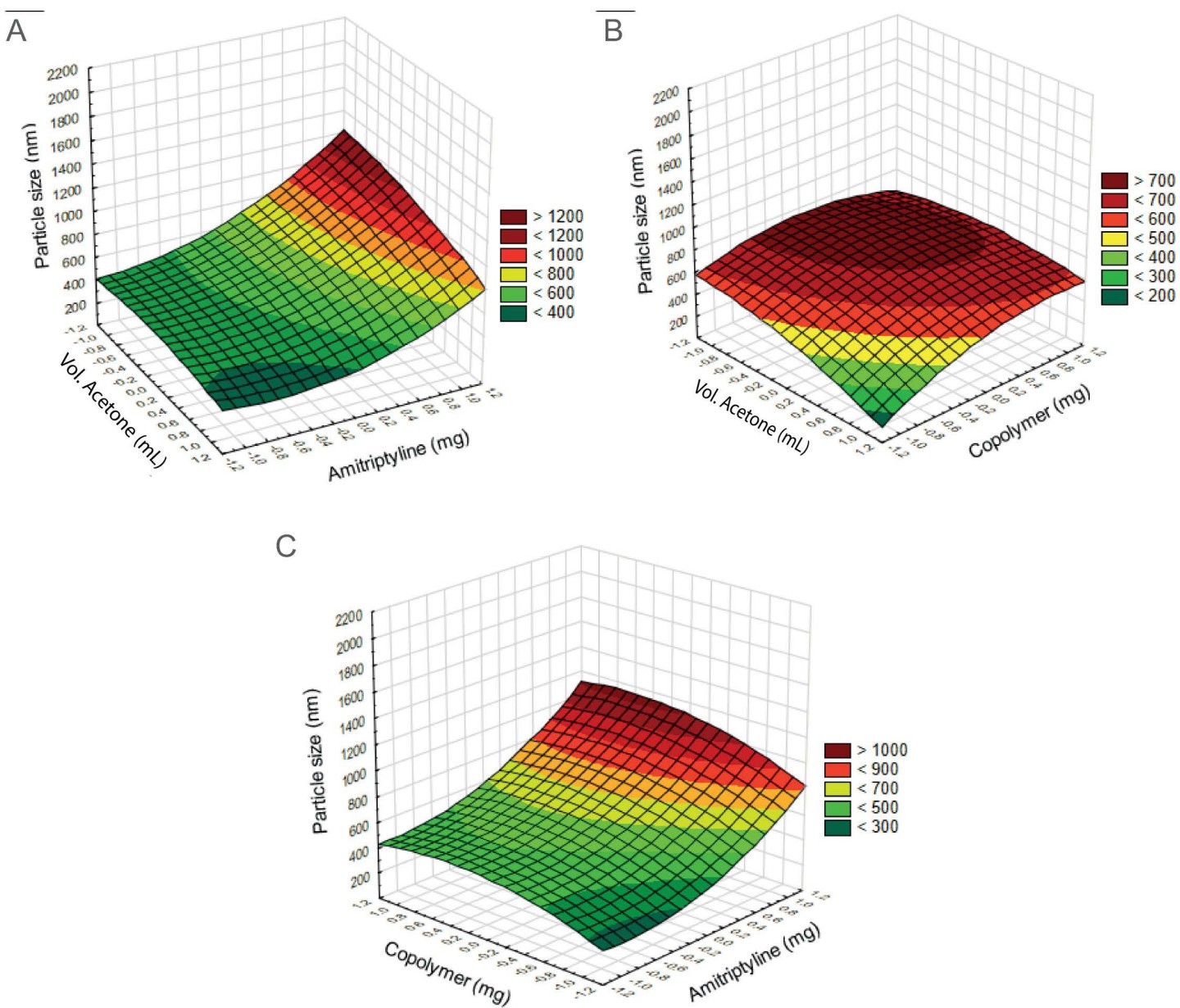

**Fig 1. Estimated response surface graph for amitriptyline nanoparticles (Ami-NPs).** A) Particle size of Ami-NPs as a function of acetone volume and amitriptyline amount of. B) Particle size of Ami-NPs influenced by acetone volume and PMVE/MA amount. C) Particle size of Ami-NPs affected by the interaction between amitriptyline and PMVE/MA amounts.

Examining the dose-response curves of Ami-S and Ami-NPs (Fig 5) revealed a notable difference in immobility reduction at the dose of 12 mg/kg. A $t$-test comparison of the absolute effects of both treatments at this dose showed that the average immobility time with Ami-NPs (118 ± 14.8 s) was significantly lower than that with Ami-S (177 ± 14.6 s; $t_{10}$ = 2.7; $p$ = 0.01; Fig 6).

## Evaluation of Ami-NPs in general locomotion and anxiety-like behavior

**Effect of Ami-NPs in the OFT.** To determine whether the reduction in immobility observed during the FST was due to an antidepressant-like response rather than a generalized

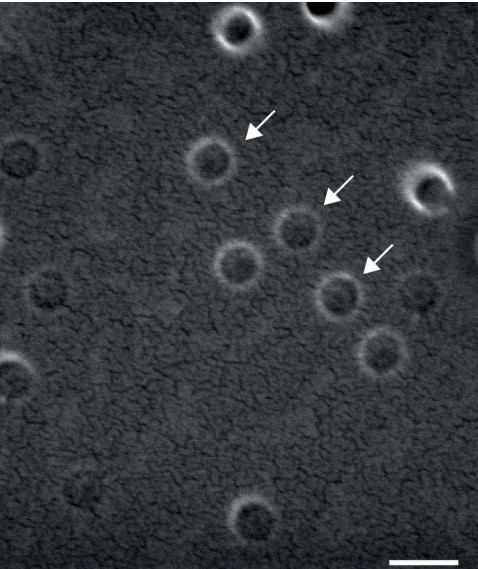

**Fig 2. Electron micrography of Ami-NPs.** Arrows indicate examples of Ami-NPs, displaying a clearly spherical shape. Scale bar = 200 nm.

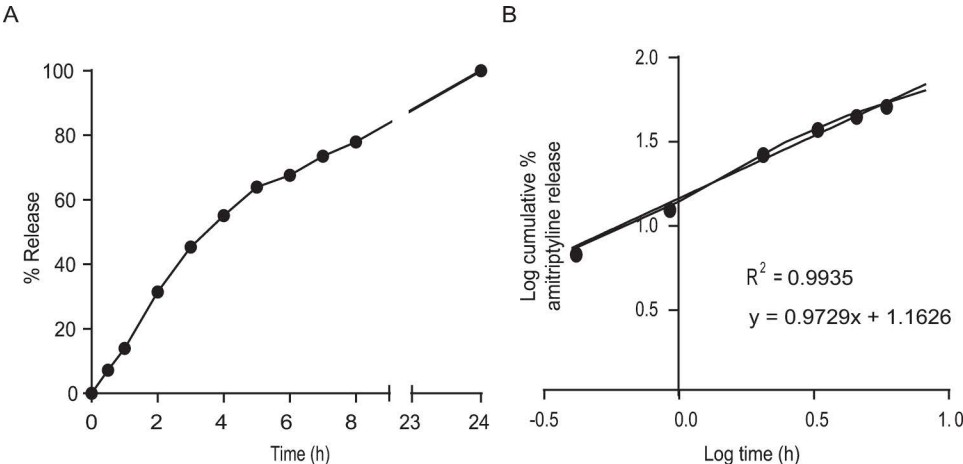

**Fig 3. *In vitro* release profile of amitriptyline from Ami-NPs.** A) Percentage of amitriptyline release over 24 h (measured at 240 nm using a UV visible spectrophotometer). B) Fitting of the Korsmeyer-Peppas model for the amitriptyline release from Ami-NPs.

**Table 3. Coefficients of determination for various dissolution kinetic models applied to the in vitro release test of amitriptyline from Ami-NPs. The Korsmeyer-Peppas model exhibited the best fit.**

| Zero order | | First order | | Higuchi | | Korsmeyer-Peppas | | |
|---|---|---|---|---|---|---|---|---|
| $R^2$ | $K_0$ | $R^2$ | $K_1$ | $R^2$ | $K_H$ | $R^2$ | $K_r$ | $n$ |
| 0.8473 | 4.5103 | 0.7368 | 0.2266 | 0.8212 | 17.707 | 0.9935 | 0.0654 | 0.9729 |

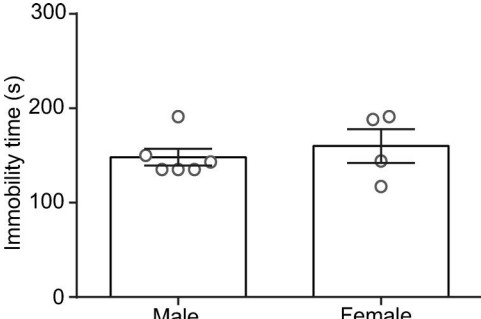

**Fig 4. Comparison of immobility time between male and female rats in the FST.** Female rats were assessed in the FST during the estrous/metestrous phases. Amitriptyline was administered at a dose of 17 mg/Kg.

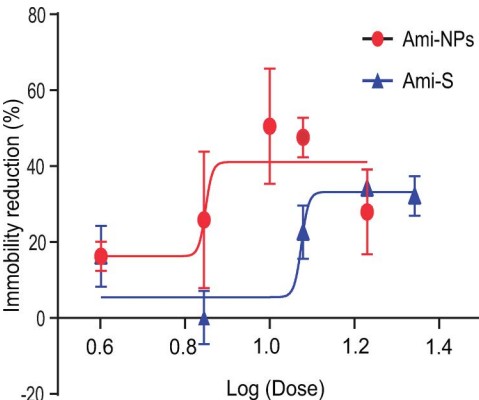

**Fig 5. Ami-NPs display enhanced potency in reducing immobility compared to amitriptyline in solution (Ami-S).** Dose-response curve for Ami-NPs (red) and Ami-S (blue). Each point represents the mean ± SEM of close to 6 animals.

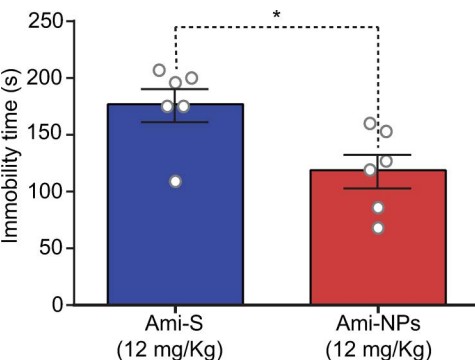

**Fig 6. Ami-NPs significantly reduce immobility compared to Ami-S.** Both Ami-NPs and Ami-S were administered at a dose of 12 mg/Kg. *, $p < 0.05$, t-test, n = 6 animals per group.

psychostimulant effect, animals were subjected to the OFT following treatment with either Ami-S or Ami-NPs at a dose of 12 mg/kg, which had shown differing effects in the FST (Fig 6).

One-way ANOVA indicated that neither Ami-S nor Ami-NPs caused significant changes in the following variables: time spent in the inner zone ($F_{[3, 20]}$ = 1.84, $p$ = 0.17; Fig 7A; S2 File), number of square crossings ($F_{[3, 20]}$ = 0.75, $p$ = 0.53; Fig 7C), number of rearings ($F_{[3, 20]}$ = 2.97, $p$ = 0.05; Fig 7D), number of self-grooming bouts ($F_{[3, 20]}$ = 0.13, $p$ = 0.93; Fig 7E), and number of fecal boluses ($F_{[3, 20]}$ = 2.16, $p$ = 0.12; Fig 7F). However, significant changes were observed in the time animals spent inactive in the OFT ($F_{[3, 19]}$ = 5.28, $p$ = 0.008; Fig 7B). *Post-hoc* Tukey's test revealed a significant increase in inactivity time ($p$ < 0.05) in animals treated with control nanoparticles (442.5 ± 35.6 s) or Ami-NPs (456.2 ± 31.1 s) compared to those treated with Ami-S (290.5 ± 20.9 s, Fig 7B).

**Effect of Ami-NPs treatment in the elevated plus maze.** One-way ANOVA analysis showed that treatment with Ami-S or Ami-NPs did not yield significant changes in the following variables: time in open arms ($F_{[3, 20]}$ = 0.6, $p$ = 0.5; Fig 8A; S3 File), time in closed arms ($F_{[3, 20]}$ = 0.98, $p$ = 0.41; Fig 8B), time in NZ ($F_{[3, 20]}$ = 0.39, $p$ = 0.17; Fig 8C), and the number of entries into open-arms ($F_{[3, 20]}$ = 0.50, $p$ = 0.68; Fig 8D). Nevertheless, treatments had a significant impact on the number of entries into closed arms ($F_{[3, 20]}$ = 8.34, $p$ = 0.0009; Fig 8E). *Post-hoc* Tukey test revealed a noteworthy decrease in closed arm entries in animals treated with Ami-NPs (2.1 ± 0.6) compared to those treated with control solution

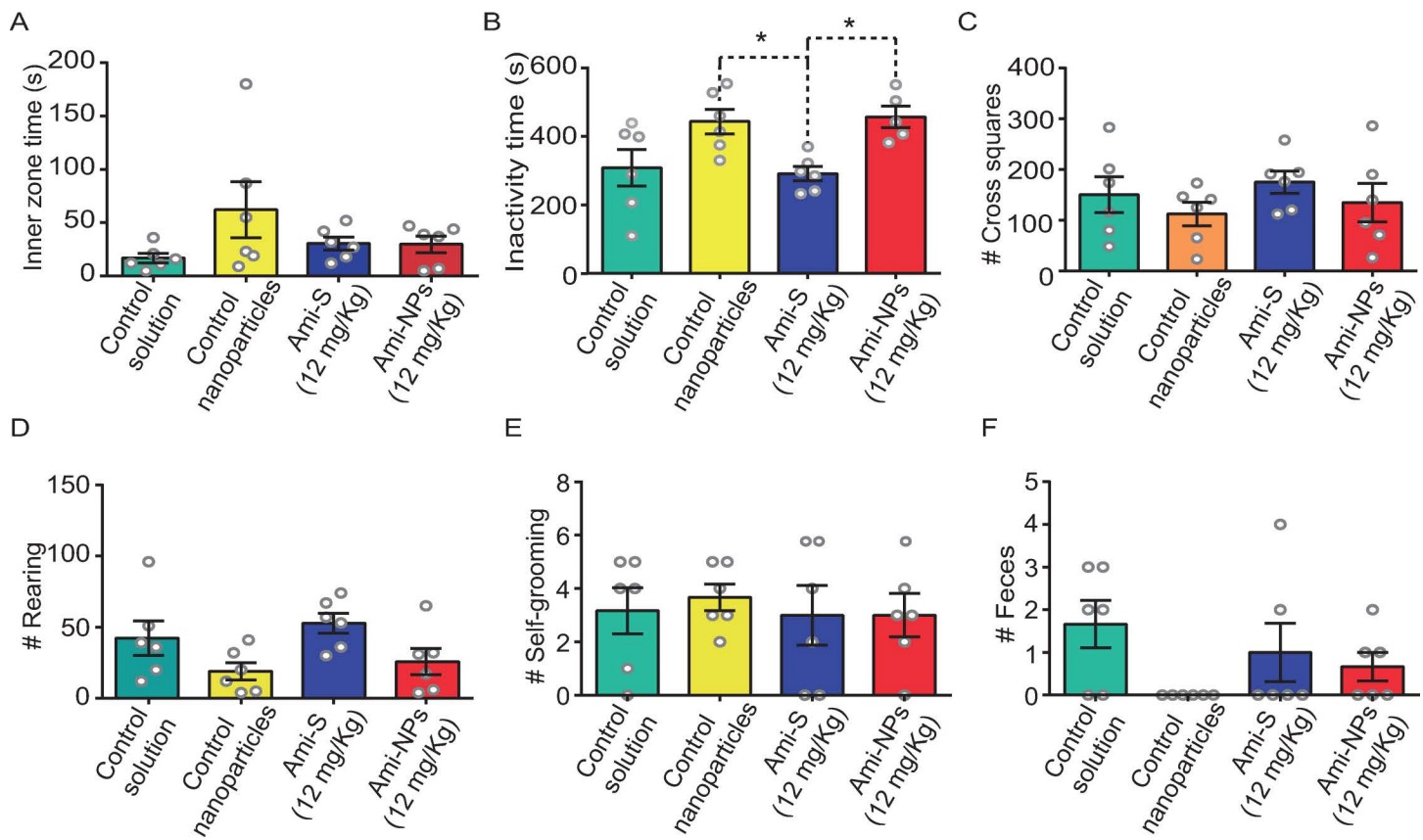

**Fig 7. General locomotion is not increased by Ami-NPs.** No significant differences were found in the Inner zone time (A), Number of square crossings (C), Number of rearings (D), Number of self-grooming bouts (E), and Number of fecal boluses (F). However, Ami-NPs exhibited a higher time of inactivity compared to Ami-S (B). *, $p$ < 0.05. ANOVA one way, Tuckey *post-hoc*; n = 6 animals per group.

(9.5 ± 1.9, $p < 0.01$), Ami-S (11.5 ± 1.6, $p < 0.001$), and control nanoparticles (8 ± 0.9, $p < 0.05$). Treatments also had a significant effect on the total number of arm entries ($F_{[3, 20]} = 5.20$, $p = 0.008$; Fig 8F). The Tukey test indicated that the total number of entries into maze arms was lower after treatment with Ami-NPs (3.3 ± 0.9) compared to Ami-S (13.5 ± 2.5, $p < 0.01$; Fig 8F).

**Effect of Ami-NPs in the MTB.** The one-way ANOVA revealed significant changes in the MBT ($F_{(3,18)} = 14.7$, $p < 0.0001$, Fig 9; S4 File). The Ami-S group showed a reduction in the number of buried marbles (3.8 ± 1.0) compared to the control solution group (12.7 ± 1.7; $p = 0.0004$; Tukey test). Similarly, the Ami-NPs group exhibited a significant decrease in the number of buried marbles compared to the control solution group (2.4 ± 0.4; $p < 0.0001$; Tukey test). Finally, animals treated with Ami-NPs buried significantly fewer marbles than the control nanoparticles group (8.7 ± 1.2; $p = 0.01$; Tukey test).

## Discussion

To the best of our knowledge, this study represents the first effort to formulate Ami-NPs with PMVE/MA and to subsequently evaluate their effects in the FST, OFT, EPM, and MBT. The primary findings indicate that animals treated with Ami-NPs showed increased efficacy

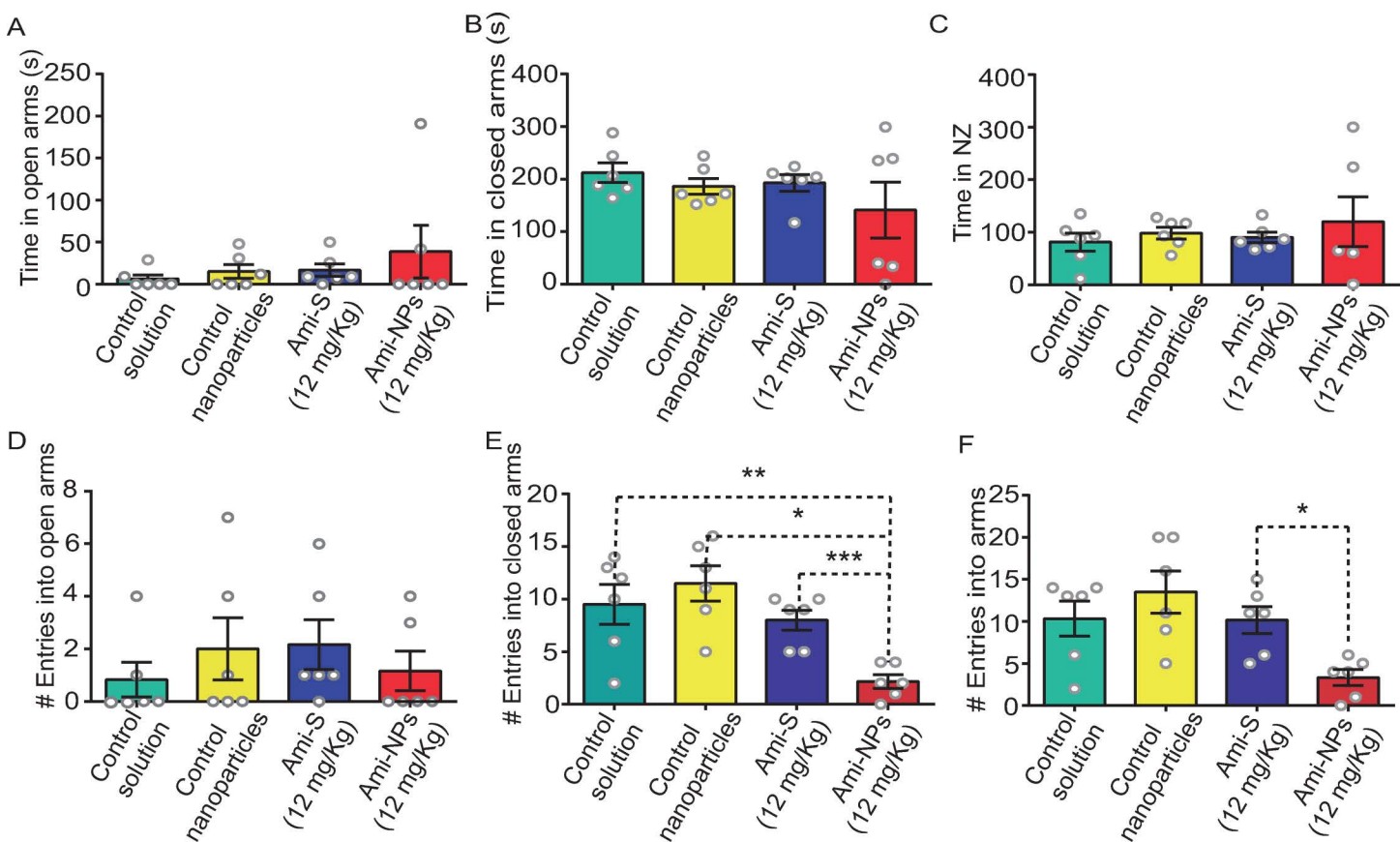

**Fig 8. Ami-NPs induce anxiolytic-like effects in the EPM.** No significant difference was found between the Ami-NPs and the other groups in terms of time spent in open arms (A), time spent in closed arms (B), time in neutral zone (NZ) (C), and number of entries into open arms (D). However, the Ami-NPs group displayed a lower number of entries into closed arms compared to all experimental groups (E). Additionally, Ami-NPs showed a reduction in the number of entries into arms compared to Ami-S (F). *, $p < 0.05$; **, $p < 0.01$; ***, $p < 0.001$. ANOVA one way, Tuckey *post-hoc*, n = 6 animals per group.

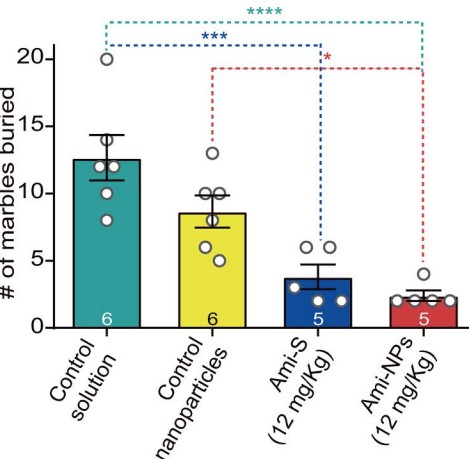

**Fig 9. Effect of Ami-NPs on the MBT.** A significant reduction in the number of buried marbles was observed in both the Ami-S and Ami-NPs (12 mg/kg each) group compared to the control solution group. Additionally, Ami-NPs showed a significant reduction in buried marbles compared to the control nanoparticles group. One-way ANOVA, Tukey post hoc: *, $p < 0.05$; ***, $p < 0.001$; ****, $p < 0.0001$. The number of animals used in each group is indicated in the columns.

compared to those injected with Ami-S in the FST. However, neither treatment led to an enhancement in general locomotion as observed in the OFT. Finally, Ami-NPs demonstrated an anxiolytic-like effect in the EPM and MBT.

## Preparation and characterization of amitriptyline nanoparticles

To prepare the Ami-NPs the nanoprecipitation method was selected for its simplicity, speed, versatility, and efficiency in producing nanometric-sized particles [44,45]. The starting materials were strategically chosen to optimize the physicochemical characteristics of Ami-NPs. The polymer was selected to ensure biodegradability and bioadhesive properties, while Poloxamer 407 was included as a stabilizer due to its ability to yield nanoparticles of reduced size. The solvent and antisolvent were chosen carefully to maintain the polymer and drug in solution and to promote controlled nanoprecipitation.

The characteristics of polymeric nanoparticles prepared by nanoprecipitation can be significantly influenced by parameters such as the polymer amount and phase proportion, which in turn impact physical attributes like size and encapsulation efficiency [46]. To achieve an optimal formulation for Ami-NPs, we implemented a design of experiments (DOE), an effective approach for optimizing formulations and minimizing product variability by adjusting various factors [47]. Specifically, we select the BBD for its efficiency and cost-effectiveness, as it requires fewer experiments compared to factorial designs [48]. Particle size was designed as primary response variable. We regarded nanoparticles size as crucial for achieving desired antidepressant-like effects of Ami-NPs. Previous studies indicated that particle larger than 200 nm are generally unable to cross the BBB [27,28]; while those under 100 nm are more prone to rapid clearance via splenic filtration [28]. However, an optimal size for polymeric nanoparticles has yet to be established. In this study, Ami-NPs were produced near the 200 nm threshold. While a further reduction in size based on BBD suggests a lower amount of Ami, this decrease could compromise the potency of Ami-NPs.

Additionally, studies indicated that using highly malleable materials in nanoparticle formulation enhances the likelihood of crossing the blood-brain barrier successfully [28]. For this reason, PMVE/MA was selected as the polymer for nanoparticles composition due to

its flexibility and bioadhesive properties, as it is commonly used to facilitate drug transport through mucosal barriers [29].

Evidence suggests that the inclusion of any surfactant improves penetration through the BBB compared to formulations without one. Specifically, Tween® 80 (polysorbate 80, P80) has been shown to enhance uptake in microglia and neurons within the hippocampus, whereas nanoparticles formulated with Poloxamer 407 (F127) demonstrate a stronger association with blood vessel structures [31]. However, in our trial the use of P80 (data not shown) resulted in nanoparticle sizes exceeding 200 nm. We also considered the possibility that Polaxamer 407 in the Ami-NPs may promote localization in blood vessel structures, potentially limiting brain targeting. Nevertheless, the surfactant coat could protect Ami from enzymatic degradation and plasma proteins, prolonging its circulation time and facilitating delivery to target structures [49].

## In vitro release studies

Ami-NPs demonstrated the ability to release Ami under *in vitro* conditions simulating physiological environments, such as the blood pH (7.4), aligning with previous reports [50,51]. Analysis the released kinetics revealed that Ami release from Ami-NPs was best described by a Korsmeyer-Peppas model (Fig 3), a model frequently applied to describe drug release from polymeric matrix systems. The Korsmeyer-Peppas model's release mechanism depends on a parameter, the "n" value. For Ami-NPs, this "n" value exceeded 0.89, indicating that Ami release followed a super case II transport [52]. This suggests that the release mechanism was primarily driven by matrix erosion and/or dissolution, allowing the drug to be released via relaxation of polymer chains. Given the nature of the PMVE/MA copolymer in Ami-NPs, dissolution in the release medium is likely in vitro. However, under in vivo conditions, matrix erosion mediated by enzymatic hydrolysis could become the main release mechanism, potentially leading to a faster release rate.

## Forced swimming test

Several behavioral tests are designed to study experimental depression based on the theory of learned helplessness [34], with the FST being one of the most widely used due to its high efficacy in identifying compounds with antidepressant-like properties, as well as its stability and practical advantages [34,41,53]. However, the FST is primarily conducted in male rodents, as evidence suggests that female rats may alter their swimming duration during phases of heightened hormonal production, such as diestrous [40,53].

In this study, we included both male and female Wistar rats, recognizing that women experience higher rates of depression than men [54]. To account for hormonal fluctuations, we compared male rats with female rats specifically in the estrous and metestrous phases, when the hormonal production is low [53], during the FST, using a saturating dose of Ami (17 mg/Kg). Since no significant differences were observed between these groups (Fig 4), we proceeded with behavioral experiments assessing the effects of Ami-NPs using male and female rats in estrous/metestrous phases of the hormonal cycle.

To evaluate the potential advantages of Ami-NPs over Ami-S, we compared their antidepressant-like effects in the FST. We conducted a DRC for each compound, following the drug injection protocol established by Porsolt *et al* [41]. Although nanoparticle delivery may involve a slower release rate [55], and may not require the same three-injections protocol used for diluted treatments, we adopt this procedure to maintain comparability with Ami-S. Conversely, this approach did not allow us to distinguish the specific effects of nanoencapsulation from those related to the injection schedule. Notably, the DRC (immobility reduction) of Ami-NPs was shifted leftward compared to the Ami-S group (Fig 5). Moreover, Ami-NPs showed a 36% reduction in the $ED_{50}$ (11.9 mg/Kg and 7 mg/Kg for Ami-S and Ami-NPs,

respectively), and approximately a 20% increase in $E_{max}$ (33.25% and 41.13% for Ami-S and Ami-NPs, respectively). Taken together, these findings suggest that nanoencapsulation enhances the antidepressant-like properties of Ami. Similar results have been reported by Margret *et al* [56], who demonstrated that nanoencapsulation of lithium carbonate with chitosan potentiated its antidepressant-like effects in both the FST and tail suspension test.

## Effect of Ami-NPs in the open field test, elevated plus maze, and marble burying test

While the FST is a reliable tool for identifying compounds with antidepressant-like properties, it should be complemented with tests evaluating locomotor activity. This precaution helps ensure that antidepressant-like effects observed in the FST are not misinterpreted as increases in general locomotion, which could lead to false positives. Our results from the OFT displayed that a saturating dose of Ami-NPs (12 mg/Kg) did not increase general locomotion (as measured by the number of square crossings) compared to other groups. However, this same dose of Ami-NPs did result in a reduction in inactivity time when compared to the control nanoparticles and Ami (12 mg/Kg). These finding support the hypothesis that Ami-NPs exhibit true antidepressant-like properties, as evidenced by opposing trends in the FST and the OFT results, consistent with clinically established antidepressant [57].

Current antidepressant medications exhibit anxiogenic properties, particularly during the initial phases of treatment, before transitioning to an anxiolytic-like effect [58]. These early anxiogenic effects contribute significantly to the discontinuation of antidepressant therapy [4]. Therefore, it is critical to identify substances that do not induce undesirable anxiogenic side effects. The findings of the present study indicate that Ami-NPs (12 mg/Kg) did not display anxiogenic-like properties compared to the control groups or the Ami-S treatment. Furthermore, the results suggest potential anxiolytic-like characteristics [58], as evidenced by the reduction in the number of closed arms entries and overall arm entries compared to the other groups analyzed. Supporting this idea, the results of the MBT suggest that Ami-NPs possess anxiolytic properties as their administration (12 mg/kg) led to a reduction in the number of buried marbles compared to both the control solution and the nanoparticle control groups. However, no significant differences were observed between Ami-S and Ami-NPs at this dose. We believe our MBT results further support the idea that Ami-NPs exhibit anxiolytic effects, as the MBT is commonly used to assess anxiety-related and obsessive-compulsive behaviour in rodents. The underlying premise of the test is that animals bury marbles in response to unfamiliar objects in their environment [59].

## Study clinical relevance

Ami is highly effective in treating major depressive disorders; however, it is not commonly used as a first-line treatment due to its side effects and the availability of newer antidepressants with fewer adverse effects. Nonetheless, Ami remains an option when newer antidepressants are ineffective or not well-tolerated (a significant percentage of patients do not respond to standard antidepressant treatments [60]. Moreover, Ami is effective in treating other conditions, such as chronic pain, migraines, insomnia. The outcomes of this research hold clinical relevance, as the nanoencapsulation of Ami enhanced its antidepressant-like effects. It is reasonable to assume that a lower dose could lead to fewer unwanted side effects. In this study, we specifically addressed the anxiogenic side effect, which is one of the leading reasons for treatment discontinuation [61]. However, additional side effects associated with Ami treatment should also be investigated in the context of Ami-NPs treatment. Further research is needed to explore the potential benefits and risks.

## Limitations of the study

This research demonstrates a method for Ami-loaded nanoparticles and provides evidence of their potentiation in antidepressant-like properties. However, it lacks a clear explanation of the mechanisms underlying the enhancement of these properties. At this stage, we can only speculate on the potential mechanisms responsible for the observed improvement. One possibility is that the Ami-loaded nanoparticles form aggregates in the blood vessels, allowing for a slower, sustained release of Ami. Another possibility is that the nanoparticles have a higher capacity of crossing the BBB. Further research is needed to explore these or other mechanisms in more detail. Additionally, evaluating the long-term effects of Ami-loaded nanoparticles would be valuable, as this study assessed the impact of acute administration.

## Conclusion

In summary, the primary outcome of this study was the successful preparation of Ami-loaded in polymeric nanoparticles. Subsequent optimization using a statistical design (BBD) was aimed at achieving an appropriate size to potentially facilitate BBB penetration. Intraperitoneal administration of these optimized nanoparticles in female Wistar rats led to a decrease in immobility during the FST, suggesting antidepressant-like properties, which were not attributable to a psychostimulant effect. Furthermore, Ami-NPs demonstrated anxiolytic-like properties, as evidenced by results from both EPM and the BMT. These findings represent an initial step toward potential future clinic applications.

## Supporting information

**S1 File.** Raw data for the dose-response curves in the FST for Ami-S and Ami-NPs.
(XLSX)

**S2 File.** Raw data for the parameters analyzed in the OFT for Ami-S, Ami-NPs, and their respective control groups.
(XLSX)

**S3 File.** Raw data for the parameter analyzed in the EPM for Ami-S, Ami-NPs, and their respective control groups.
(XLSX)

**S4 File.** Raw data for Ami-S, Ami-NPs, and their respective control groups in the MBT.
(XLSX)

## Author contributions

**Conceptualization:** Javier Alamilla, Ramon Eduardo Valadez-Lemus, Néstor Mendoza-Muñoz.

**Data curation:** Ramon Eduardo Valadez-Lemus, Juana María Jiménez-Vargas.

**Formal analysis:** Javier Alamilla, Jose L. Góngora-Alfaro.

**Funding acquisition:** Néstor Mendoza-Muñoz.

**Investigation:** Ramon Eduardo Valadez-Lemus.

**Methodology:** Javier Alamilla, Jose L. Góngora-Alfaro, Juana María Jiménez-Vargas.

**Resources:** Javier Alamilla.

**Supervision:** Javier Alamilla, Jose L. Góngora-Alfaro.

**Validation:** Juana María Jiménez-Vargas, Néstor Mendoza-Muñoz.

**Writing – original draft:** Ramon Eduardo Valadez-Lemus.

**Writing – review & editing:** Javier Alamilla, Néstor Mendoza-Muñoz.

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
