## [Decision Letter · Decision Letter 0]

26 Aug 2024

PONE-D-24-23966Nanoencapsulation of Amitriptyline enhances the potency of antidepressant-like effects and mitigates certain anxiogenic-like effects in Wistar rats.PLOS ONE

Dear Dr. Alamilla,

Thank you for submitting your manuscript to PLOS ONE. After careful consideration, we feel that it has merit but does not fully meet PLOS ONE’s publication criteria as it currently stands. Therefore, we invite you to submit a revised version of the manuscript that addresses the points raised during the review process.

We look forward to receiving your revised manuscript.

Kind regards,

Vara Prasad Saka

Academic Editor

PLOS ONE

Journal requirements: 1. When submitting your revision, we need you to address these additional requirements. Please ensure that your manuscript meets PLOS ONE's style requirements, including those for file naming. The PLOS ONE style templates can be found at https://journals.plos.org/plosone/s/file?id=wjVg/PLOSOne_formatting_sample_main_body.pdf and https://journals.plos.org/plosone/s/file?id=ba62/PLOSOne_formatting_sample_title_authors_affiliations.pdf. 2. We note that the grant information you provided in the ‘Funding Information’ and ‘Financial Disclosure’ sections do not match.  When you resubmit, please ensure that you provide the correct grant numbers for the awards you received for your study in the ‘Funding Information’ section. 3. Thank you for stating the following financial disclosure:  [CONACYT-SEP funding under Grant CB A1-S-39237.].  Please state what role the funders took in the study.  If the funders had no role, please state: ""The funders had no role in study design, data collection and analysis, decision to publish, or preparation of the manuscript."" If this statement is not correct you must amend it as needed. Please include this amended Role of Funder statement in your cover letter; we will change the online submission form on your behalf. 4. Please expand the acronym “CONACYT-SEP” (as indicated in your financial disclosure) so that it states the name of your funders in full.This information should be included in your cover letter; we will change the online submission form on your behalf. 5. Thank you for stating the following in the Acknowledgments Section of your manuscript: [The authors appreciate the technical assistance of Manuel Aguilar Franco (CFTA-UNAM) for the SEM. Ramon Valadez-Lemus acknowledge to Consejo Nacional de Ciencia y Tecnología (CONACYT) by the grant number 928892.]We note that you have provided funding information that is not currently declared in your Funding Statement. However, funding information should not appear in the Acknowledgments section or other areas of your manuscript. We will only publish funding information present in the Funding Statement section of the online submission form. Please remove any funding-related text from the manuscript and let us know how you would like to update your Funding Statement. Currently, your Funding Statement reads as follows:   [CONACYT-SEP funding under Grant CB A1-S-39237.]. Please include your amended statements within your cover letter; we will change the online submission form on your behalf. 6. Your ethics statement should only appear in the Methods section of your manuscript. If your ethics statement is written in any section besides the Methods, please move it to the Methods section and delete it from any other section. Please ensure that your ethics statement is included in your manuscript, as the ethics statement entered into the online submission form will not be published alongside your manuscript.  7. We note that Figure 2 in your submission contain copyrighted images. All PLOS content is published under the Creative Commons Attribution License (CC BY 4.0), which means that the manuscript, images, and Supporting Information files will be freely available online, and any third party is permitted to access, download, copy, distribute, and use these materials in any way, even commercially, with proper attribution. For more information, see our copyright guidelines: http://journals.plos.org/plosone/s/licenses-and-copyright. We require you to either (1) present written permission from the copyright holder to publish these figures specifically under the CC BY 4.0 license, or (2) remove the figures from your submission: a. You may seek permission from the original copyright holder of Figure 2 to publish the content specifically under the CC BY 4.0 license.  We recommend that you contact the original copyright holder with the Content Permission Form (http://journals.plos.org/plosone/s/file?id=7c09/content-permission-form.pdf) and the following text:“I request permission for the open-access journal PLOS ONE to publish XXX under the Creative Commons Attribution License (CCAL) CC BY 4.0 (http://creativecommons.org/licenses/by/4.0/). Please be aware that this license allows unrestricted use and distribution, even commercially, by third parties. Please reply and provide explicit written permission to publish XXX under a CC BY license and complete the attached form.” Please upload the completed Content Permission Form or other proof of granted permissions as an ""Other"" file with your submission.  In the figure caption of the copyrighted figure, please include the following text: “Reprinted from [ref] under a CC BY license, with permission from [name of publisher], original copyright [original copyright year].” b. If you are unable to obtain permission from the original copyright holder to publish these figures under the CC BY 4.0 license or if the copyright holder’s requirements are incompatible with the CC BY 4.0 license, please either i) remove the figure or ii) supply a replacement figure that complies with the CC BY 4.0 license. Please check copyright information on all replacement figures and update the figure caption with source information. If applicable, please specify in the figure caption text when a figure is similar but not identical to the original image and is therefore for illustrative purposes only. 8. Please include your tables as part of your main manuscript and remove the individual files. Please note that supplementary tables (should remain/ be uploaded) as separate ""supporting information"" files".

Reviewers' comments:

Reviewer's Responses to Questions

**Comments to the Author**

1. Is the manuscript technically sound, and do the data support the conclusions?

Reviewer #1: Yes

Reviewer #2: Yes

2. Has the statistical analysis been performed appropriately and rigorously? 

Reviewer #1: Yes

Reviewer #2: Yes

3. Have the authors made all data underlying the findings in their manuscript fully available?

Reviewer #1: Yes

Reviewer #2: Yes

4. Is the manuscript presented in an intelligible fashion and written in standard English?

Reviewer #1: Yes

Reviewer #2: No

5. Review Comments to the Author

Reviewer #1: Publishable article subject to minor technical corrections in language and referencing.

Authors are requested to follow the referencing style of the journal. Methods and statistical plan are as per expectations.

The need for the research and the implications, take-aways must figure in the concluding part of the introductory

section.

Reviewer #2: The manuscript presents a well-designed study investigating the potential benefits of nanoformulating amitriptyline (Ami). The authors effectively demonstrate that Ami nanoparticles (Ami-NPs) exhibit enhanced potency and efficacy in reducing immobility in the forced swimming test (FST) compared to Ami in solution. However, the study could be strengthened by addressing the following critical questions:

Overall Critical Questions

Anxiogenic-like Effects: While the authors report that Ami-NPs mitigate certain anxiogenic-like effects in the elevated plus-maze, they do not provide a comprehensive analysis of anxiety-related behaviors. It would be valuable to include data from additional anxiety-like tests (e.g., light-dark box, Vogel test) to establish a more robust understanding of the anxiolytic properties of Ami-NPs.

Mechanism of Action: The study provides evidence for enhanced bioavailability of Ami-NPs, but it lacks a deeper exploration of the underlying mechanisms. Future studies could investigate whether the nanoparticles enhance Ami's absorption, distribution, or metabolism, and how these factors contribute to the observed effects.

Long-Term Effects: The authors focus on acute administration of Ami-NPs and Ami-S. It would be interesting to evaluate the long-term effects of these formulations, including potential tolerance or dependence issues.

Clinical Relevance: While the preclinical findings are promising, it is essential to address the clinical relevance of these results. How might the development of Ami-NPs translate into improved patient outcomes? Could this approach potentially reduce the incidence of side effects associated with Ami therapy?

Characterization of Ami-NPs: While the authors provide basic characterization data (size, polydispersity index, zeta potential, encapsulation efficiency), additional information on the stability of the nanoparticles over time and under different storage conditions would be helpful.

Introduction Analysis

Literature review: While the introduction provides some relevant references, a more comprehensive literature review could be included to strengthen the argument for nanoencapsulation as a promising approach.

Specificity: The discussion of the benefits of nanoencapsulation could be more specific, addressing how it might improve amitriptyline's bioavailability, reduce side effects, and enhance its efficacy.

Comparison with other approaches: The introduction could briefly discuss other potential approaches to improving amitriptyline's efficacy, such as combination therapy or alternative antidepressants, to highlight the unique advantages of nanoencapsulation.

Methodology Analysis

Justification for specific methods: The rationale behind the choice of specific nanoparticles components (PMVE/MA, Poloxamer 407) and the Box-Behnken design (BBD) for optimization could be further elaborated on.

Detailed explanation of analytical methods: While the methodology mentions characterization techniques (DLS, SEM, HPLC), a more detailed explanation of how each method is used to assess particle size, morphology, and encapsulation efficiency would be beneficial.

In vitro release assay: Additional information on the rationale behind the chosen parameters, such as the selection of PBS (pH 7.4) and the duration of the experiment, could be provided.

Animal studies: The justification for using different doses of Ami-S and Ami-NPs in the FST could be clarified. Including details on estrous cycle determination methods strengthens the research.

6. PLOS authors have the option to publish the peer review history of their article (what does this mean? ). If published, this will include your full peer review and any attached files.

**Do you want your identity to be public for this peer review?** For information about this choice, including consent withdrawal, please see our Privacy Policy .

Reviewer #1: **Yes: ** Dr. Raja Chakraverty

Reviewer #2: **Yes: ** Bharat Kumar Reddy Sanapalli

---

## [Author Response · Author response to Decision Letter 1]

6 Nov 2024

November 6, 2024

Dear Editor and Reviewers,

We would like to express our gratitude for the time and effort dedicated to improving this manuscript. We have addressed all of your questions and suggestions, which we believe have substantially improved the manuscript. Briefly, the main changes are as follows:

- We conducted new experiments using the Marble Buring test to asses anxiety-like behaviour in rodents.

- We create a new figure (Fig. 9) to illustrate these new results.

- All sections of the manuscript have been revised to incorporate these findings.

- The English language has been carefully edited to correct any grammatical or spelling errors.

- The introduction section was revised to highlight nanoparticles as a promising pharmacological strategy.

- The Method section was expanded to provide further details on the methodological approaches, including the evaluation of the nanoparticles, identification of estrous phases in the rat cytology, and rationale behind the dose-response curve for each compound analyzed.

- Tables were included into the main text as required.

- The Discussion section was revised to reflect Reviewer # 2 suggestions.

- The title of the study has been changed, as we consider the new results provide evidence supporting this modification.

- Additional supporting material containing the raw data has been uploaded to reinforce our findings.

Bellow, you will find our responses to your specific questions (highlighted in green). Please note that the page and line numbers correspond to the clean version of the manuscript, titled “Manuscript Valadez-Lemus”.

[CONACYT-SEP funding under Grant CB A1-S-39237.].

4. Please expand the acronym “CONACYT-SEP” (as indicated in your financial disclosure) so that it states the name of your funders in full.

[The authors appreciate the technical assistance of Manuel Aguilar Franco (CFTA-UNAM) for the SEM. Ramon Valadez-Lemus acknowledge to Consejo Nacional de Ciencia y Tecnología (CONACYT) by the grant number 928892.]

R: Thank you for this information. In this part the student wanted to thank CONACYT for a doctoral fellowship. We eliminated the fellowship’s ID to avoid any misunderstanding.

[CONACYT-SEP funding under Grant CB A1-S-39237.].

R: This information as been updated in the cover letter to the Editor

R: We eliminate the ethics statement from the manuscript, and we maintained it only the Methods section. Thank you.

7. We note that Figure 2 in your submission contain copyrighted images. All PLOS content is published under the Creative Commons Attribution License (CC BY 4.0), which means that the manuscript, images, and Supporting Information files will be freely available online, and any third party is permitted to access, download, copy, distribute, and use these materials in any way, even commercially, with proper attribution. For more information, see our copyright guidelines: http://journals.plos.org/plosone/s/licenses-and-copyright.

R: We apologize for the misunderstanding. The Figure 2 is an original photograph captured from imaging the nanoencapsulation of amitriptyline using an electronic microscope. The image was later edited using Adobe Illustrator (arrows and scale bar). This image has not been published or presented in any proceeding of local or international meetings. If necessary, we can provide the original, unedited image.

8. Please include your tables as part of your main manuscript and remove the individual files. Please note that supplementary tables (should remain/ be uploaded) as separate ""supporting information"" files".

R: Done. Thank you.

Reviewers' comments:

Reviewer's Responses to Questions

Comments to the Author

1. Is the manuscript technically sound, and do the data support the conclusions?

Reviewer #1: Yes

Reviewer #2: Yes

2. Has the statistical analysis been performed appropriately and rigorously?

Reviewer #1: Yes

Reviewer #2: Yes

3. Have the authors made all data underlying the findings in their manuscript fully available?

Reviewer #1: Yes

Reviewer #2: Yes

4. Is the manuscript presented in an intelligible fashion and written in standard English?

Reviewer #1: Yes

Reviewer #2: No

R: Thank you very much for your comment. We have conducted a thorough review of the manuscript’s writing, making several modifications have been made, as shown the tracked-changes document. We hope these revisions have improved the quality of the text.

5. Review Comments to the Author

Reviewer #1: Publishable article subject to minor technical corrections in language and referencing.

Authors are requested to follow the referencing style of the journal.

R: References have been revised and formatted to the journal’s requirements.

Methods and statistical plan are as per expectations.

R: Thak you very much

The need for the research and the implications, take-aways must figure in the concluding part of the introductory section.

R: Thank you very much for your helpful suggestions. We have added lines outlining the state of the art on amitriptyline nanoparticles and the need for this study (Page: 4, lines: 86-91). Additionally, a new paragraph has been included to state the study’s aims and key takeaways (Page: 5, lines: 110-118).

Reviewer #2: The manuscript presents a well-designed study investigating the potential benefits of nanoformulating amitriptyline (Ami). The authors effectively demonstrate that Ami nanoparticles (Ami-NPs) exhibit enhanced potency and efficacy in reducing immobility in the forced swimming test (FST) compared to Ami in solution.

R: Thank you very much for your kind comments.

However, the study could be strengthened by addressing the following critical questions:

Overall Critical Questions

Anxiogenic-like Effects: While the authors report that Ami-NPs mitigate certain anxiogenic-like effects in the elevated plus-maze, they do not provide a comprehensive analysis of anxiety-related behaviors. It would be valuable to include data from additional anxiety-like tests (e.g., light-dark box, Vogel test) to establish a more robust understanding of the anxiolytic properties of Ami-NPs.

R: We appreciate this suggestion. We conducted additional experiments using the marble burying test (MBT), a method used to asses general anxiety and obsessive-compulsive behavior (Kedia & Chattarji, 2014). The results showed a reduction in the number of buried marbles in animals treated with Ami-NPs compared to the control nanoparticles group, suggesting anxiolytic-like properties of Ami-NPs. These results have been added to the Results section (Page 18 and 19: lines: 395-409), and Figure 9. Additionally, the Methods, Discussion and Abstract sections have been updated to reflect these new results.

Mechanism of Action: The study provides evidence for enhanced bioavailability of Ami-NPs, but it lacks a deeper exploration of the underlying mechanisms. Future studies could investigate whether the nanoparticles enhance Ami's absorption, distribution, or metabolism, and how these factors contribute to the observed effects.

Long-Term Effects: The authors focus on acute administration of Ami-NPs and Ami-S. It would be interesting to evaluate the long-term effects of these formulations, including potential tolerance or dependence issues.

R: We agree with the reviewer’s insightful perspective. We have added a new subsection in the Discussion titled “Limitation of the study”, addressing the lack of clarity regarding the mechanisms of action of the Ami-loaded nanoparticles. This section discusses the increased potency of the antidepressant-like properties and emphasizes the need of further research to better understand the mechanisms of action of Ami-NPs and their long-term effects, as the results analyzed in this study were acute (page: 24, Paragraph: 541-549).

Clinical Relevance: While the preclinical findings are promising, it is essential to address the clinical relevance of these results. How might the development of Ami-NPs translate into improved patient outcomes? Could this approach potentially reduce the incidence of side effects associated with Ami therapy?

R: We appreciate this comment. We have added a new paragraph in the Discussion section, titled “Study clinical relevance”. In that section, we discuss the potential for translating our results to clinical settings and the possibility of a reducing side effects through the use of lower doses of amitriptyline (Page: 24, lines: 528-539).

Characterization of Ami-NPs: While the authors provide basic characterization data (size, polydispersity index, zeta potential, encapsulation efficiency), additional information on the stability of the nanoparticles over time and under different storage conditions would be helpful.

R: We have added lines in the Results section detailing the stability of the nanoparticles formulation (Page: 15, lines: 325-327).

Introduction Analysis

Literature review: While the introduction provides some relevant references, a more comprehensive literature review could be included to strengthen the argument for nanoencapsulation as a promising approach.

Specificity: The discussion of the benefits of nanoencapsulation could be more specific, addressing how it might improve amitriptyline's bioavailability, reduce side effects, and enhance its efficacy.

Comparison with other approaches: The introduction could briefly discuss other potential approaches to improving amitriptyline's efficacy, such as combination therapy or alternative antidepressants, to highlight the unique advantages of nanoencapsulation.

R: A new paragraph has been added to the introduction section to strengthen the argument for nanoencapsulation as a promising approach: (Page: 13, lines: 67-84)

Methodology Analysis

Justification for specific methods: The rationale behind the choice of specific nanoparticles components (PMVE/MA, Poloxamer 407) and the Box-Behnken design (BBD) for optimization could be further elaborated on.

R: The section of “Preparation and characterization of Amitriptyline nanoparticles”, in the Discussion section has been expanded to explain the rationale behind the election of nanoparticle components (Page: 19, lines

---

## [Decision Letter · Decision Letter 1]

11 Dec 2024

Nanoencapsulation of Amitriptyline enhances the potency of antidepressant-like effects and exhibits anxiolytic-like effects in Wistar rats.

PONE-D-24-23966R1

Dear Dr. Alamilla,

We’re pleased to inform you that your manuscript has been judged scientifically suitable for publication and will be formally accepted for publication once it meets all outstanding technical requirements.

Kind regards,

Vara Prasad Saka

Academic Editor

PLOS ONE

Additional Editor Comments (optional):

Reviewers' comments:

Reviewer's Responses to Questions

**Comments to the Author**

1. If the authors have adequately addressed your comments raised in a previous round of review and you feel that this manuscript is now acceptable for publication, you may indicate that here to bypass the “Comments to the Author” section, enter your conflict of interest statement in the “Confidential to Editor” section, and submit your "Accept" recommendation.

Reviewer #1: All comments have been addressed

Reviewer #2: All comments have been addressed

2. Is the manuscript technically sound, and do the data support the conclusions?

Reviewer #1: Yes

Reviewer #2: Yes

3. Has the statistical analysis been performed appropriately and rigorously? 

Reviewer #1: Yes

Reviewer #2: Yes

4. Have the authors made all data underlying the findings in their manuscript fully available?

Reviewer #1: Yes

Reviewer #2: Yes

5. Is the manuscript presented in an intelligible fashion and written in standard English?

Reviewer #1: Yes

Reviewer #2: Yes

6. Review Comments to the Author

Reviewer #1: Publishable and in order. Thank you for your efforts in improving the quality of the manuscript. Get spelling typo checks just once.

Reviewer #2: I'm satisfied with the revisions and appreciate the authors' efforts and dedication greatly, thank you.

7. PLOS authors have the option to publish the peer review history of their article (what does this mean? ). If published, this will include your full peer review and any attached files.

**Do you want your identity to be public for this peer review?** For information about this choice, including consent withdrawal, please see our Privacy Policy .

Reviewer #1: **Yes: ** Dr Raja Chakraverty

Reviewer #2: No

---

## [Editor Report · Acceptance letter]

PONE-D-24-23966R1

PLOS ONE

Dear Dr. Alamilla,

I'm pleased to inform you that your manuscript has been deemed suitable for publication in PLOS ONE. Congratulations! Your manuscript is now being handed over to our production team.

Kind regards,

on behalf of

Dr. Vara Prasad Saka

Academic Editor

PLOS ONE